# Auto-Generating Weak Labels for Real & Synthetic Data to Improve Label-Scarce Medical Image Segmentation

**Tanvi Deshpande**[1]                   TANVIMD@STANFORD.EDU
[1] *Stanford University*

**Eva Prakash**[1]                    EPRAKASH@STANFORD.EDU
**Elsie Gyang Ross**[1]                 ELSIE.ROSS@STANFORD.EDU
**Curtis Langlotz**[1]                  LANGLOTZ@STANFORD.EDU
**Andrew Ng**[1]                      ANG@STANFORD.EDU
**Jeya Maria Jose Valanarasu**[1]           JMJOSE@STANFORD.EDU

**Editors:** Accepted for publication at MIDL 2024

## Abstract

The high cost of creating pixel-by-pixel gold-standard labels, limited expert availability, and presence of diverse tasks make it challenging to generate segmentation labels to train deep learning models for medical imaging tasks. In this work, we present a new approach to overcome the hurdle of costly medical image labeling by leveraging foundation models like Segment Anything Model (SAM) and its medical alternate MedSAM. Our pipeline has the ability to generate *weak labels* for any unlabeled medical image and subsequently use it to augment label-scarce datasets. We perform this by leveraging a model trained on a few gold-standard labels and using it to intelligently prompt MedSAM for weak label generation. This automation eliminates the manual prompting step in MedSAM, creating a streamlined process for generating labels for both real and synthetic images, regardless of quantity. We conduct experiments on label-scarce settings for multiple tasks pertaining to modalities ranging from ultrasound, dermatology, and X-rays to demonstrate the usefulness of our pipeline. The code is available at github.com/stanfordmlgroup/Auto-Generate-WLs/.
**Keywords:** Label Scarcity, Weak Labeling, Segmentation, Synthetic Data

## 1. Introduction

The process of automatically identifying and delineating specific structures within medical images, i.e. segmentation, holds immense use-cases in various tasks, including diagnosis, treatment planning, and surgical procedures. Deep learning-based methods such as (Ronneberger et al., 2015; Zhou et al., 2018; Milletari et al., 2016; Valanarasu et al., 2020, 2021; Chen et al., 2021; Tang et al., 2022; Ma et al., 2024) have recently constituted advancements in medical image segmentation. Most of these methods are fully supervised networks and need a good amount of data and labels for them to perform well.

While it is typical in computer vision tasks to obtain many segmentation labels through crowd workers, medical imaging tasks require experts to annotate the images. Due to the high cost of obtaining gold-standard labels from qualified medical professionals, practitioners often encounter the issue of *label scarcity* in many medical imaging tasks. Annotating medical images for segmentation is also particularly difficult due to the fine-grained nature

of the label. In contrast, however, unlabeled data is more freely available. In addition, generative models like GANs (Goodfellow et al., 2020) and diffusion models (Kazerouni et al., 2023) can help generate synthetic medical images, further boosting the data size. However, this does not help augment datasets in a fully supervised pipeline, as we also need paired labels with the images to train the model.

Recently, Kirillov et al. (2023) released Segment Anything Model (SAM), a vision foundation model trained on over 11 million images and 1 billion masks. It has the unique ability to segment any image out-of-the-box in a zero-shot setting. Segmentation using SAM can be done automatically without any inputs. However, this typically does not lead to optimal performance, and so segmentation is usually done with the help of prompting techniques on the image like points or boxes (Kirillov et al., 2023). As SAM was trained on natural images, it is often found to be suboptimal when applied to medical imaging tasks (de Oliveira et al., 2023). To address this gap, Ma et al. (2023) introduced MedSAM, a fine-tuned version of SAM created specifically for the medical imaging domain, trained on over 1 million images across 15 modalities. MedSAM was found to be superior in terms of performance on a variety of medical image segmentation tasks when compared to SAM.

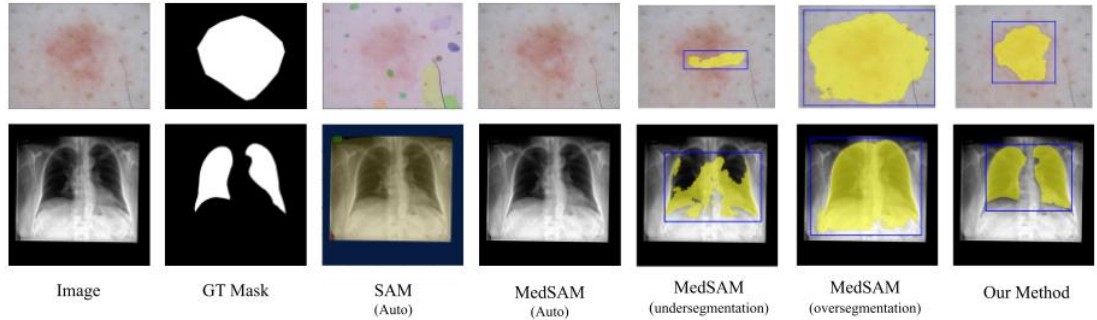

Figure 1: Labels obtained from different configurations of SAM and MedSAM. Our method auto-generates effective input prompts (bounding boxes) using only very few annotations to generate high quality weak labels while SAM and MedSAM fail in auto options and are sensitive to input prompts in manual option.

Intuitively, MedSAM should help us annotate the unlabeled images out-of-the-box and obtain labels to augment our training dataset. However, one major limitation of SAM and MedSAM is their sensitivity to input prompts. Both models are only as good as the point or box inputs they receive on a given image. In specific, for medical images, the area of interest to be segmented may be particularly subtle within the image with thin fine-grained boundaries. Also, inputs that include too much of the target or background result in over- or undersegmentation, requiring precise input prompts to get a good segmentation. This can be seen in Fig. 1, where we show labels generated from SAM and MedSAM. MedSAM clearly gives us better labels than SAM, but it is very sensitive to the bounding box prompt we use and is still constrained by the manual prompting part. We also note that auto-prompting MedSAM performs poorly and often leads to blank segmentation predictions. Therefore, selecting an appropriate input prompt is key to ensuring success. Also, eliminating the need

for manual prompting could allow practitioners to auto-generate labels for any number of unlabeled real or synthetic data, which would be very useful in label-scarce scenarios.

To this end, we present a new pipeline that tackles the challenge of limited labeled data by harnessing the power of foundation models like MedSAM. Our approach leverages coarse labels, generated by training segmentation models on few labels (ranging from 25 to 50), to guide the selection of inputs for unlabeled data fed into the SAM model. This process effectively creates a richer dataset, enabling the training of a significantly more accurate model while using only a few gold-standard labels and eliminating the need for time-consuming, expensive manual labeling. This enriched dataset fuels the training of deep learning models with improvements in the dice accuracy ranging from 6.6% to 72.3% for medical image segmentation datasets like BUSI, ISIC, and CANDID-PTX.

In summary, our contributions are as follows:

- We introduce a pipeline that automatically generates efficient weak labels for any unlabeled data using MedSAM, eliminating the need for manual prompts.

- We show that when using these weak labels to augment training datasets, we observe notable performance gains in deep learning models. This is especially impactful in scenarios with limited labeled data.

- We conduct thorough validation of the method and ablation studies across ultrasound (BUSI), dermoscopy (ISIC), and X-ray (CANDID-PTX) datasets.

## 2. Related Works

After the introduction of SAM (Kirillov et al., 2023) and MedSAM (Ma et al., 2023), these vision foundation models have been directly applied to a myriad of medical imaging tasks, such as brain tumor segmentation (Peivandi et al., 2023), eye feature segmentation (Maquiling et al., 2023), liver tumor segmentation, and lung nodule segmentation (He et al., 2023), in both fine-tuning and zero-shot settings.

Several approaches have also been introduced tailoring SAM for specific medical imaging tasks. For example, Lin et al. (2023) tailors SAM specifically for ultrasound segmentation by injecting features into SAM's encoder through a parallel CNN branch. Shin et al. (2023) jointly trains a network on heterogeneous ultrasound datasets using condition embedding blocks along with SAM, to allow SAM to adapt to each dataset separately. Chen et al. (2023) adapts SAM for 3D medical image segmentation, by incorporating 3D adapters into SAM's encoder.

There has also been work involving self-prompting SAM for medical image segmentation. One branch of such work involves separately learning prompts for SAM and combining them with SAM's existing architecture. One such approach includes learning a pixel-wise classifier from SAM's own embedding space and encoder to prompt SAM in few-shot settings (Wu et al., 2024). In addition, Lei et al. (2023) use extreme points from 3D medical images to generate 2D bounding-box prompts for SAM. Pandey et al. (2023) first train YOLOv8 on several image-mask pairs to detect bounding-boxes for the regions to segment, which are then fed to SAM. Lastly, Anand et al. (2023) use localization to a template image to prompt SAM in one-shot settings. The other branch of such work involves altering SAM's

architecture to incorporate learning prompts. Shaharabany et al. (2023) learn prompts for SAM by training a separate prompt encoder to automatically generate prompts, rather than conditioning on manual prompts, as SAM does. Cui et al. (2023) use SAM to generate weak segmentation labels, which are then used to fine-tune SAM.

Annotations generated from SAM can also be used to augment existing segmentation pipelines, such as U-Nets (Zhang et al., 2023). In addition, there has been work regarding sampling prompts from SAM (Qi et al., 2023), and SAM has been used for a variety of non-medical tasks, such as inpainting (Yu et al., 2023) and captioning (Wang et al., 2023).

## 3. Method

We propose a pipeline that automatically generates weak labels for both unlabeled real and synthetic data using MedSAM, eliminating the need for manual prompting. Our pipeline includes training a small model on the available gold-standard labels, using predictions from this model to generate prompts for MedSAM, and retraining using a larger dataset consisting of the gold-standard and weak-labeled data. Our method is illustrated in Fig. 2.

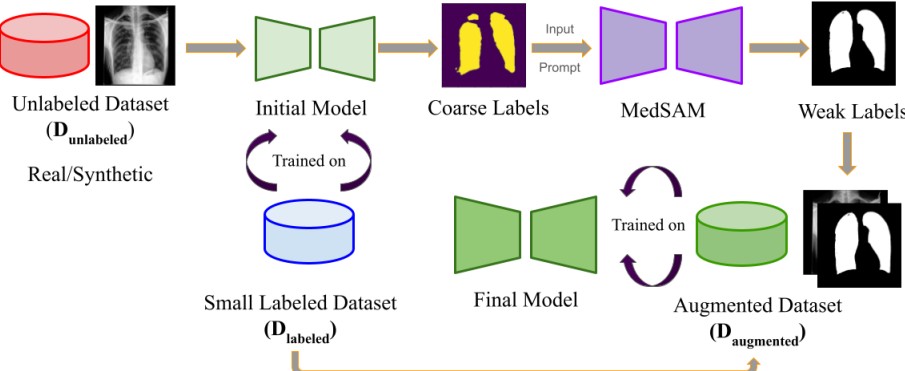

Figure 2: An illustration of our pipeline to auto-generate weak labels for unlabeled data. We use the limited annotations to train an initial model that generates low-quality coarse labels on unlabeled data. We then select inputs from these coarse labels as prompts to MedSAM to create higher-quality weak labels. These weak labels are used to train a stronger segmentation model.

**Preliminaries** A label-scarce scenario in a medical imaging setting can be defined as a task for which there exists a small amount of labeled (gold-standard) data, and a larger pool of unlabeled data. Let the small labeled dataset be denoted by $\mathcal{D}_{labeled}$, where we assume less than 50 gold-standard labels from medical practitioners. We chose this number after considering some real clinical scenarios, in which we encountered situations where it was difficult to obtain more than 50 gold standard labels. We acknowledge that this number may vary depending on the specific task, and therefore, we conduct ablation studies using different label quantities in Section 5. Notably, acquiring 50 high-quality labels is typically more feasible than the hundreds to thousands often required for traditional segmentation models. We define the larger pool of unlabeled data as $\mathcal{D}_{unlabeled}$, which may comprise real medical scans or synthetic images generated by models like diffusion models. The evaluation set, denoted as $\mathcal{D}_{test}$, remains separate for performance assessment.

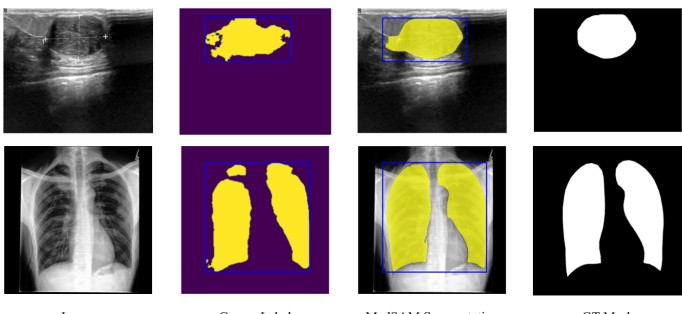

| Image | Coarse Label | MedSAM Segmentation | GT Mask |

Figure 3: Example of prompting and labeling from coarse label.

We also define *weak labels* as labels that are not gold-standard (as they are not created by experts), but are still useful in improving the model performance.

**Coarse Label Generation**   To make use of the full potential of limited annotations, we first train an initial model $\theta$ on the labeled dataset $\mathcal{D}_{labeled}$. This model is used to generate coarse labels for the unlabeled dataset $\mathcal{D}_{unlabeled}$. These coarse labels, despite potentially limited accuracy due to the model's training on a smaller number of labels, serve as valuable inputs for automatically prompting MedSAM to produce higher-quality weak labels.

**Input Selection**   The prompts to MedSAM can be either points or bounding-boxes. Both "positive points" (corresponding to portions of the image to segment) and "negative points" (corresponding to portions of the image *not* to segment) can be given as input point prompts. For bounding-boxes, one needs to draw a box around their region of interest and feed that as the prompt input to MedSAM. Although these prompts are an easier alternative to pixel-by-pixel annotations, they still require manual effort and a good knowledge about the segmentation task.

To auto-generate labels using MedSAM without any manual intervention, we first pick up the coarse label prediction from the initial model $\theta$. If the coarse label is empty, we filter out the sample, since there is no basis for a prompt to MedSAM. If not, we pick the largest contiguous blocks from the coarse label and filter out small blocks, as those are likely to be noise. For point inputs, we use the middle of the largest contiguous region(s) of the coarse label as the prompt to MedSAM; in addition, low-probability points from the original prediction mask are used as "negative points" for the MedSAM model. For bounding-box inputs, we compute the minimum and maximum indices of the largest contiguous region(s) of the coarse label as the prompt to MedSAM. An example of the input-selection process is shown in Fig. 3. We fixed these techniques after thorough experimentation on various input selection methods (App. C). While the input selection technique we describe works well for multiple tasks, we do acknowledge that for some tasks, there could be more intelligent prompting techniques that could perform better. However, we would like to point out that our input selection technique is much more generic and can be helpful in obtaining useful weak labels. It should also be noted that these weak labels are far better than the ones generated by SAM and MedSAM through automatic option (App B).

**Labeling and Filtering**   To ensure the quality of weak labels generated from unlabeled images, we implement a filtering process that eliminates extreme cases. Specifically, masks

containing a disproportionately high percentage of pixels ($> 97\%$) belonging to either the background or the segmented class are excluded. This filtering step safeguards against misleading or uninformative masks that could potentially hinder model performance.

**Augmenting the training data** Finally, the weak labels generated for unlabeled images are combined with $\mathcal{D}_{labeled}$ to form the final *augmented* dataset $\mathcal{D}_{augmented}$, consisting of images, where a small number of labels (25-50) are gold-standard, and the rest are weak labels. This augmented dataset will be used in the final training of the model.

**Synthetic Data** In cases where data scarcity might further exacerbate label scarcity, we explore the potential of synthetic image generation as a means to potentially increase dataset size. Thus, we additionally generate synthetic images from each dataset in order to evaluate the performance of our pipeline on such data. We train a denoising diffusion probabilistic model (DDPM) with a UNet backbone following (Ho et al., 2020). We set the hyperparameters following (Wolleb et al., 2021), notably with a learning rate of $1e^{-4}$, image size of 256, and batch size of 4.

## 4. Experiments and Results

In this section, we give details about the datasets we use, experimental setup, and the results that validate the usefulness of our method.

### 4.1. Datasets

We utilize four datasets featuring 2D binary medical image segmentation tasks. Within each dataset, we randomly select $N$ random samples from the training set to be the "gold-standard" labels. Note that we experiment on different values of $N$ to mimic different label-scarce settings. The rest of the training set is considered unlabeled data (i.e. we do not use the ground-truth labels in our experiments, but generate weak labels for these images). This approach enables us to train and evaluate our model without relying on the full set of ground-truth annotations.

**BUSI:** The Breast Ultrasound Images Dataset (BUSI) consists of ultrasound images in three classes: benign, malignant, and normal (Al-Dhabyani et al., 2020). We focus on segmentation of benign and malignant lesions from the ultrasound images. The number of samples with segmentation masks is 650. We use a randomized train-test split of 80–20, resulting in 130 samples in the test set.

**ISIC:** The International Skin Imaging Collaboration (ISIC) dataset consists of dermoscopic lesion segmentation tasks (Gutman et al., 2016). It comprises 1279 dermoscopic lesion images, divided into a training set of 900 images and a test set of 379 images.

**CANDID:** We use a subset of the CANDID-PTX dataset (Feng et al., 2021), consisting of 500 images, which consists of binary segmentation of lungs from chest X-rays. We use a randomized 80–20 train-test split.

### 4.2. Implementation Details

We use UNet++ (Zhou et al., 2018) as our base segmentation network. Please note that our method is agnostic to the choice of network; we picked UNet++ as it is stable in training

on label-scarce conditions. We use a combination of DICE loss and binary cross-entropy loss with a scaling ratio of 1 and 0.5 respectively to train our models. We use the SGD optimizer with a learning rate of $10^{-3}$, and a cosine annealing learning rate scheduler with minimum learning rate $10^{-5}$. All images were also reshaped to a resolution of $256 \times 256$. We use a batch size of 4 and train for 100 epochs on NVIDIA A4000 GPUs.

## 4.3. Results

We provide the quantitative results for 3 datasets in Table 1 in label-scarce settings. The performance of models trained for datasets augmented with weak labels is compared against the performance of a model trained on just the base dataset $\mathcal{D}_{labeled}$, which consists of gold-standard (GS) labels. In addition, we compare against SAM's automatic option as well as UniverSeg (Butoi et al., 2023), using $\mathcal{D}_{labeled}$ as the support dataset. We provide results for both bounding-box and point prompts (automatically generated by our pipeline), observing that different prompts are more suited to different datasets. We achieve improvements of up to 73.3%, with more dramatic improvements where the initial DICE score was lower; though the weak labels may not be an exact match with the gold-standard labels, they are accurate enough to provide a boost in performance. More qualitative results can be seen in the appendix (App. A).

Table 1: Results using our method in label-scarce settings. GS = Gold Standard.

| # GS Labels | # Weak Labels | DICE | | | | | | | | |
|---|---|---|---|---|---|---|---|---|---|---|
| | | BUSI | | | ISIC | | | CANDID-PTX | | |
| UniverSeg | | 0.3681 | | | 0.5257 | | | 0.7700 | | |
| | | Auto | Box | Points | Auto | Box | Points | Auto | Box | Points |
| 25 | 0 | 0.3059 | 0.3059 | 0.3059 | 0.6123 | 0.6123 | 0.6123 | 0.8182 | 0.8182 | 0.8182 |
| 25 | 25 | 0.2629 | 0.3613 | 0.3777 | 0.5810 | 0.7367 | 0.7884 | 0.6396 | 0.8879 | 0.8726 |
| 25 | 50 | 0.2401 | 0.4326 | 0.4661 | 0.5907 | 0.7587 | 0.8087 | 0.5519 | 0.9044 | 0.8872 |
| 25 | 100 | 0.2124 | 0.4661 | **0.5302** | 0.5893 | 0.7424 | **0.8483** | 0.4115 | **0.9096** | 0.8443 |

**Synthetic Data** In addition to experiments using real data, we conduct experiments using data from a diffusion model trained on unlabeled samples from each of the datasets (Table 2). We train the diffusion model on the train splits of BUSI, ISIC, and CXR-COVID (Fraiwan et al., 2023) datasets respectively and sample it multiple times to generate the required number of synthetic images. As diffusion models can produce large amounts of synthetic data after being trained on just a few samples, the number of possible weak labels is higher.

## 4.4. Ablation Studies

In addition to our main experiments, we perform ablation studies examining various elements of our pipeline and also discuss some limitations in this section.

**Label scarce settings** We vary the number of images in the base dataset to observe the effect on the performance boost from weak labeling and present the results in Table 3. Though increasing gold-standard labels is expected to reduce performance gains from

Table 2: Synthetic Data Experiments

| # GS Labels | # Weak Labels | DICE | | |
| --- | --- | --- | --- | --- |
| | | BUSI | ISIC | CANDID-PTX |
| 25 | 0 | 0.3059 | 0.6123 | 0.8182 |
| 25 | 50 | 0.4177 | 0.7762 | 0.8997 |
| 25 | 250 | 0.4312 | 0.7228 | 0.9073 |
| 25 | 500 | 0.4301 | 0.7247 | 0.9022 |

weak labels, when starting with only 10 base images, the quality of coarse labels plummets, rendering MedSAM prompting ineffective in such extreme label scarcity. However, it can be noted that we always gain performance while using our pipeline.

Table 3: Ablation on label-scarce settings. Experiments are conducted on BUSI dataset.

| # GS Labels | # Weak Labels (Syn. Data) | DICE (GS) | DICE (GS + WL) |
| --- | --- | --- | --- |
| 10 | 100 | 0.2327 | 0.2637 |
| 25 | 100 | 0.3059 | 0.4661 |
| 50 | 100 | 0.4379 | 0.5416 |
| 100 | 100 | 0.5575 | 0.6155 |

**SAM vs. MedSAM**   We ablate on the use of the base SAM model compared to the fine-tuned MedSAM model (Table 4) and observe that MedSAM outperforms SAM.

Table 4: SAM vs. MedSAM. Experiments are conducted on BUSI dataset.

| Model | # GS Labels | # Weak Labels | DICE |
| --- | --- | --- | --- |
| — | 25 | 0 | 0.3059 |
| SAM | 25 | 100 | 0.3886 |
| MedSAM | 25 | 100 | 0.4661 |

**Limitations**   Our work focuses on 2D ultrasound, X-ray, and dermoscopic data. We recognize that tasks involving highly intricate structures may require different input selection approaches due to potential sensitivities in our method (App. D). We also did not touch on 3D segmentation tasks; future work could investigate extensions to address such scenarios.

## 5. Conclusion

We introduce a new method for addressing label-scarce scenarios in medical image segmentation using recent advancements in vision foundation models. By selecting inputs to MedSAM from coarse labels trained on a small gold-standard dataset, we create augmented datasets with weak labels that can be auto-generated for any number of unlabeled data. Using these augmented datasets, we train models that obtain significant boosts in performance on label-scarce settings. Weak labels generated through our method can also be used to improve human-in-the-loop annotation processes.

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

## Appendix A. Qualitative Results

In this section, we provide some examples comparing predictions from the base model trained on 25 gold-standard labels, compared to the final model trained on augmented datasets consisting of 25 gold-standard and 100 weak labels. We observe that the final prediction is far more accurate than the base model predictions.

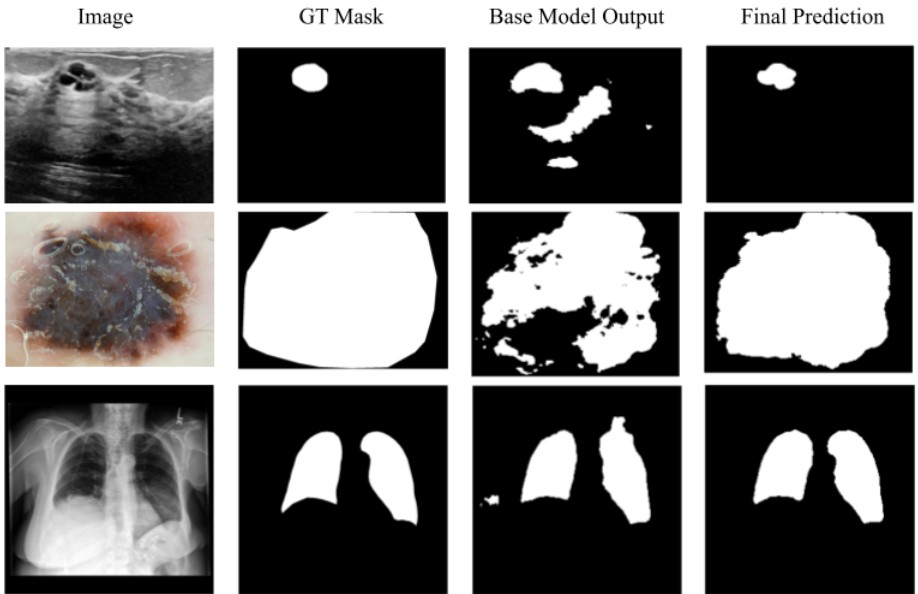

Figure 4: Example outputs from each dataset

## Appendix B. Comparison to SAM and MedSAM Automatic Segmentation

In this section, we compare our weak labels to the outputs from SAM and MedSAM in their automatic options (Fig. 5). SAM tends to segment many regions, in which it is not clear which is the target. In contrast, MedSAM tends to generate blank masks in its auto option. The weak labels generated from our method is better and is closer to GT while also auto-generated.

## Appendix C. Input Selection

We provide comparisons of various input selection methods to our method in Fig. 6. In the third column, for example, we choose inputs based on the darkest pixels present in the image. In the fourth column, we use a bounding box based solely on the image size rather than the image content. In the fifth and sixth columns, we include outputs from our pipeline, using point and box prompts that were automatically generated from the base model.

Our pipeline outperforms both of these alternative methods. In addition, we note that our pipeline is dataset-agnostic; it may not be the case that the target region contains the

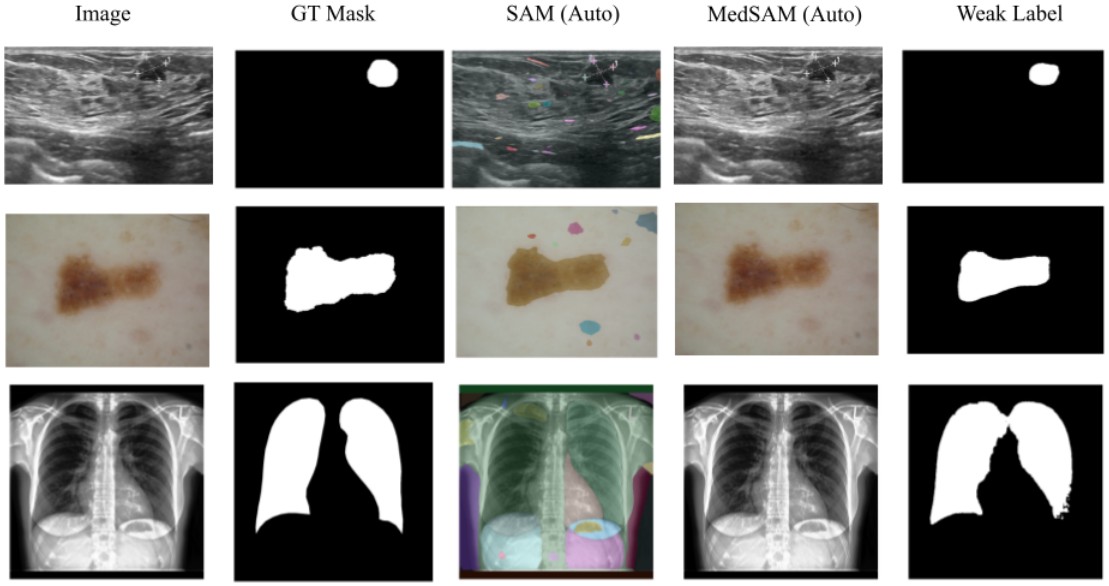

Figure 5: Comparison to SAM and MedSAM automatic segmentation for each dataset

darkest pixels in an image, for example, or that it can be captured with a bounding box encompassing most of the image.

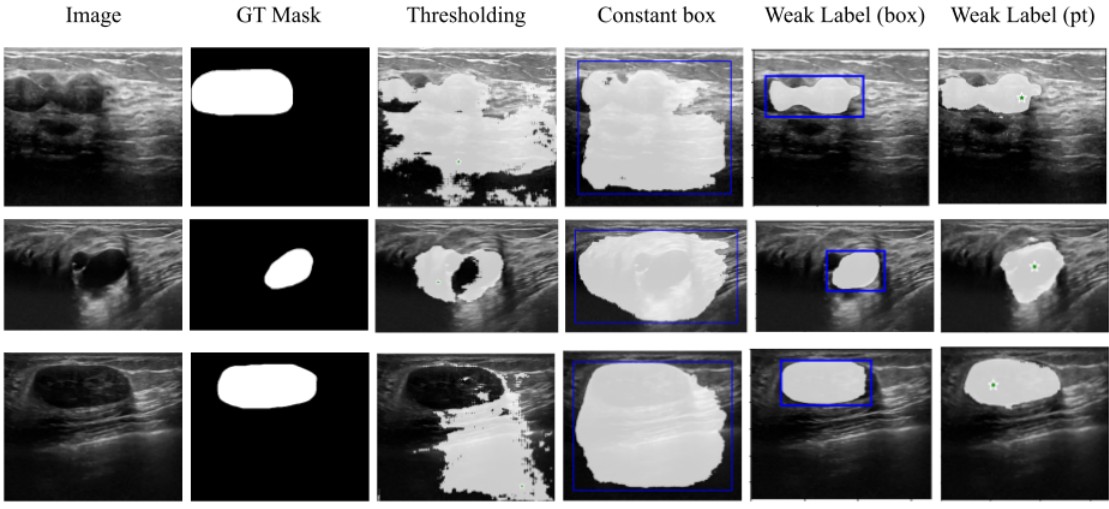

Figure 6: Comparison of input selection methods

## Appendix D. Limitations: Results on CHASE Dataset

We provide results on the CHASE dataset (Fraz et al., 2012), which consists of 28 images for the task of retinal vessel segmentation, using a train-test split of 20 and 8 images, respectively.

We note that our pipeline does not improve performance on the CHASE dataset; segmentation of particularly fine-grained vessels may require alternate input selection methods due to the increased sensitivity MedSAM may have to inputs on them. Furthermore, we note that due to the extremely small size of the CHASE dataset (28 images), the label-scarcity of the gold-standard dataset may be too extreme to provide informative coarse labels to help us generate high-quality weak labels.

Table 5: CHASE

| # GS Labels | # Weak Labels | DICE |
|:---:|:---:|:---:|
| 5 | 0 | 0.2406 |
| 5 | 5 | 0.2557 |
| 5 | 10 | 0.218 |
| 5 | 15 | 0.2081 |

