# OpenReview forum: "Auto-Generating Weak Labels for Real & Synthetic Data to Improve Label-Scarce Medical Image Segmentation"
_MIDL.io/2024/Conference — MIDL 2024 Poster_

### Official Review · Reviewer_FF7P · 2024-02-26

**Confidence:** 4
**Preliminary Rating:** 3
**Recommendation:** Poster
**Final Rating:** 2

**Summary:**

This paper proposed a method that leverages foundation models to generate weak labels and then augment label-scarce datasets. The validation experiments on three public datasets demonstrated the outperformance of the proposed method.

**Strengths:**

The idea of leveraging the foundation model is interesting and attractive.
The experiment covering four datasets is comprehensive.
The presentation is easy to follow, and the figure is clear to understand.

**Weaknesses:**

The innovation of this paper is the proposed pipeline, which is quite tricky and not strong enough.
Why do not use MedSAM in the first-round training as the initial model? What is the advantage of using the initial model compared with using MedSAM to generate coarse labels?

**Detailed Comments:**

The author needs to prove that the result of the proposed method is better than fine-tuning SAM to directly generate labels for the augmented datasets.

**Justification Of Final Rating:**

Thanks to the authors for the careful response. However, the innovation and content are still not strong enough for me to be accepted by MIDL. A more necessary motivation is expected, and more experiments are required.

**Justification Of The Preliminary Rating:**

The proposed pipeline is interesting but simple, which does not meet my expectations. However, the presentation is easy to follow, and the experiment is comprehensive, which outperforms other papers I have been reviewing.

**Questions To Address In The Rebuttal:**

Please clarify the disadvantage of fine-tuning MedSAM to generate coarse labels compared with training the initial model.

**Special Issue:**

No

---

> ### Author Response · Authors · 2024-03-17
>
> **1) Why not use MedSAM in the first-round training as the initial model?**
>
> As demonstrated in our paper, MedSAM has no auto option; generating predictions on images without a prompt largely results in blank masks. While we could use MedSAM to generate coarse labels independent of our pipeline, doing so would require manual prompts (again from, for instance, medical experts), making it labor-intensive. Hence, our method of generating prompts automatically is advantageous, as it has no manual intervention, and so could be very fast to generate weak labels for any number of real or synthetic data. We have added baseline results from the automatic version of SAM in our paper; without prompts, SAM segments many regions of the image, in which it is not clear which corresponds to the target, and we select a random region as the target mask. Training using these weak labels results in performance drops for all three datasets, in comparison to the notable gains from using our method.

---

> > ### Author Response · Authors · 2024-03-26
> >
> > We believe that we have thoroughly addressed the reviewer's concerns and kindly request the reviewer to let us know if there are  any additional issues following our response. We thank the reviewer for their time.

---

### Official Review · Reviewer_Jv3k · 2024-02-28

**Confidence:** 5
**Preliminary Rating:** 2
**Recommendation:** Poster

**Summary:**

The paper presents an auto-prompting approach to generate pseudo labels for semi-supervised segmentation training applied to 2D medical image segmentation. Multiple 2D image modalities and segmentation tasks were evaluated and comparison of SAM and MedSAM approaches to generate the pseudo labels was performed.

**Strengths:**

* The idea of auto-generating prompts to improve the consistency of generated pseudo labels from MedSAM or SAM based methods is interesting and potentially meaningful when these methods are used for improving accuracy of downstream classifiers.
* Paper considers evaluation of multiple 2D imaging modalities and evaluated impact of input prompting on the generated pseudo labels.

**Weaknesses:**

* Besides the generation of auto-prompts that basically perform simple morphological operations to extract bounding box on the MedSAM segmentations, the method is rather incremental.
* Whereas the stated problem is the variability of generated pseudo labels due to prompts, a detailed analysis of variability in the generated pseudo labels due to different methods and to what extent they impact downstream accuracy is missing. Most of the analysis focused on the number of training examples used to train the initial model, which seems to have a minimal impact on the accuracy.
* Baseline comparison to other methods such as UniverSeg are missing.

**Detailed Comments:**

Please refer to comments and suggestions in strengths and weaknesses.

**Justification Of The Preliminary Rating:**

The idea of generating auto-prompts to generate pseudo labels is conceptually novel, albeit the method itself is rather incremental. Moreover, the evaluations are limited and comparison to other baselines are not considered.

**Questions To Address In The Rebuttal:**

* Whereas the stated problem is the variability of generated pseudo labels due to prompts, a detailed analysis of variability in the generated pseudo labels due to different methods and to what extent they impact downstream accuracy is missing. Most of the analysis focused on the number of training examples used to train the initial model, which seems to have a minimal impact on the accuracy.
* Baseline comparison to other methods such as UniverSeg are missing.

**Special Issue:**

No

---

> ### Author Response · Authors · 2024-03-17
>
> **1) Addressing the comment on methodology being incremental**
>
> Please note that our work is on the Validation & Application track, which is intended for “well-validated applications of deep learning algorithms in medical imaging”. Our focus was on validating that foundation models can be used effectively in real-life label-scarce scenarios (which was shown through experiments on a variety of modalities and dataset sizes). We do propose a new pipeline and show that it is generalizable. While it is possible to develop specific heuristic-based methods (such as thresholding based on image contours or darkness) for each dataset, our focus was on creating a dataset- and modality-agnostic pipeline. The generalizability of our method can be noted from the performance gains on all datasets without changes in the method.
>
> **2) Variability of generated pseudo labels**
>
> Figures 1 and 5 qualitatively demonstrate how variable MedSAM and SAM are in generating pseudo labels. To further motivate this claim, we have now done experiments involving using SAM’s auto option, which is included in the main results table; without prompts, SAM segments many regions of the image; selecting one arbitrarily to use as the segmentation label results in performance drops.
>
> We also ablated on the type of auto-prompt we chose for our method in Appendix C. We provided comparisons of various input selection methods to our method in Figure 6.
>
> **3) Comparisons with UniverSeg**
>
> We have added this to our main results table, along with baseline results corresponding to labels from SAM in its auto option.  For UniverSeg, we evaluate on the test set using the gold-standard dataset as the support images and labels; our method outperforms UniverSeg on all 3 datasets. Without prompts, SAM segments many regions of the image, in which it is not clear which corresponds to the target, and we select a random region as the target mask. Training using these weak labels results in performance drops for all three datasets, in comparison to the notable gains from using our method. We include the table here.
>
> | \# GS Labels | \# Weak Labels | |  |  |  | **DICE** |  |  |  |  |
> |--------------|---------------|----------|----------|----------|----------|----------|----------|----------|----------|----------|
> |              |               |  | **BUSI** |  |  | **ISIC** | |  | **CANDID-PTX** |  |
> | UniverSeg    |               |   | 0.3681   |   |   | 0.5257   |   |   | 0.7700     |     |
> |              |                        | Auto     | Box      | Points   | Auto     | Box      | Points   | Auto       | Box        | Points     |
> | 25           | 0             | 0.3059   | 0.3059   | 0.3059   | 0.6123   | 0.6123   | 0.6123   | 0.8182   | 0.8182     | 0.8182     |
> | 25           | 25            | 0.2629   | 0.3613   | 0.3777   | 0.5810   | 0.7367   | 0.7884   | 0.6396   | 0.8879     | 0.8726     |
> | 25           | 50            | 0.2401   | 0.4326   | 0.4661   | 0.5907   | 0.7587   | 0.8087   | 0.5519   | 0.9044     | 0.8872     |
> | 25           | 100           | 0.2124   | 0.4661   | **0.5302**| 0.5893   | 0.7424   | **0.8483**| 0.4115   | **0.9096**| 0.8443     |

---

> > ### Author Response · Authors · 2024-03-26
> >
> > We believe that we have thoroughly addressed the reviewer's concerns and kindly request the reviewer to let us know if there are  any additional issues following our response. We thank the reviewer for their time.

---

### Official Review · Reviewer_V1WH · 2024-03-04

**Confidence:** 3
**Preliminary Rating:** 2
**Final Rating:** 2

**Summary:**

This work addresses the challenge of generating segmentation labels for medical imaging tasks. By leveraging foundation models like Segment Anything Model (SAM) and its medically adapted counterpart, MedSAM, the authors propose a pipeline capable of generating weak labels for unlabeled medical images, thereby augmenting datasets with limited labels. This method involves training a model on a few gold-standard labels, which then guides MedSAM in generating weak labels, streamlining the process for both real and synthetic images. Experiments across various medical imaging modalities, including ultrasound, dermatology, and X-rays, demonstrate the effectiveness of this approach, with notable improvements in model performance using the augmented datasets.

**Strengths:**

The paper is well-structured and clearly written. It provides a comprehensive review of related work. This work addresses a very important challenge in the field and is fresh in it's attempt to leverage tools such as MedSAM, the latest advancements from computer vision world for medical image segmentation.

**Weaknesses:**

One potential weakness is the reliance on foundation models like SAM and MedSAM, which, despite their adaptability, may still not capture the nuances of highly specialized medical imaging tasks.

Also, this work does not provide a comparison of the state-of-the-art segmentation tools from the respective modalities to proposed method.

**Detailed Comments:**

All comments have been provided over the other sections of the review.

**Justification Of Final Rating:**

I am retaining my original rating for the paper. While I do appreciate the efforts of the authors for rebuttal, I do not believe that sufficient additions have been made to the paper to meet the required quality.

**Justification Of The Preliminary Rating:**

This method addresses a significant challenge in the field, presenting a novel solution with the potential to advance research and application in medical image analysis. The comprehensive experimentation across various modalities adds credibility to the claims and demonstrates the method's versatility.

However, the lack of comparison with state-of-the-art segmentation tools, the limited discussion on the application to 3D imaging tasks, and the potential generalizability issues of foundation models in specialized medical imaging scenarios are notable weaknesses. These limitations suggest areas where the paper could be strengthened through additional analysis, experimentation, and discussion.

**Questions To Address In The Rebuttal:**

How do the state-of-the-art, domain specific models correspond to being trained with the weak labels generated by the proposed method?

---

> ### Author Response · Authors · 2024-03-17
>
> **1) On Reliance of SAM and MedSAM:**
>
> Please note that our work is not focused on solving the nuances that occur in highly specialized medical imaging tasks. Our main goal is to show how label-scarce tasks in medical imaging can be boosted using weak labels to augment training datasets; we observe notable performance gains (up to 73%). We would like to point out that our method being generalizable and adaptable by relying on MedSAM is in fact a strength and not a weakness.
>
> **2) Comparison with domain-specific models:**
>
> Our work is designed to be implemented on top of any state-of-the-art, domain-specific model. In our work, we considered UNet++ as our base network, as most domain-specific segmentation frameworks follow a UNet/UNet++ network architecture with domain-specific loss functions or modules. Our contribution is in boosting the size of the available dataset for training, which can be implemented on top of any domain-specific segmentation model.

---

> > ### Author Response · Authors · 2024-03-26
> >
> > We believe that we have thoroughly addressed the reviewer's concerns and kindly request the reviewer to let us know if there are  any additional issues following our response. We thank the reviewer for their time.

---

### Author Response · Authors · 2024-03-18

We thank all of the reviewers for their feedback. We have addressed all of their comments in the following replies and made changes accordingly in the revised edition of the paper.

---

### Meta-Review · Area_Chair_xXbS · 2024-04-04

**Recommendation:** Accept (Poster)
**Confidence:** 3

**Metareview:**

Reviewers appreciated the concept of the paper of using foundation models to generate labels, but they all raised initial concerns about this paper
- Benchmarking is insufficient with respect to well-performing baselines
- Insufficient analysis of the differences between pseudolabels generated by the different approaches
- Generalizability of foundation models to specific medical imaging tasks is not clear

The authors engaged in the rebuttal and modified significantly their paper, in my opinion addressing all concerns raised in the relatively short reviews satisfactorily.

Thus, while reviewers maintained or lowered their scores, I recommend acceptance.

---

### Decision · Program_Chairs · 2024-04-06

Accept (Poster)